# REVIVE AND RECOUPLE: MITIGATING PLASTICITY LOSS IN TRANSFORMER ARCHITECTURES

## ABSTRACT

A key trait of general intelligence is the ability to continuously adapt and learn in non-stationary environments. Neural networks progressively lose their ability to learn in such settings. This phenomenon is known as plasticity loss. Existing mitigation techniques for plasticity loss either require extensive additional memory, compute, or suffer from loss of crucial task information, causing a performance drop. Crucially, while this phenomenon is well-studied in traditional MLPs, there is a significant lack of insight regarding the loss of plasticity in transformer architectures. In this work, we find that plasticity loss also occurs in transformer architectures, both in dense layers and layer norm parameters. To address this, we present a novel two-step framework Revive And Recouple (RnR) designed to mitigate plasticity loss while preserving crucial knowledge, thereby avoiding performance drops. Our experiments show that RnR significantly outperforms current approaches on transformer architectures in Continual Learning (CL) scenarios.

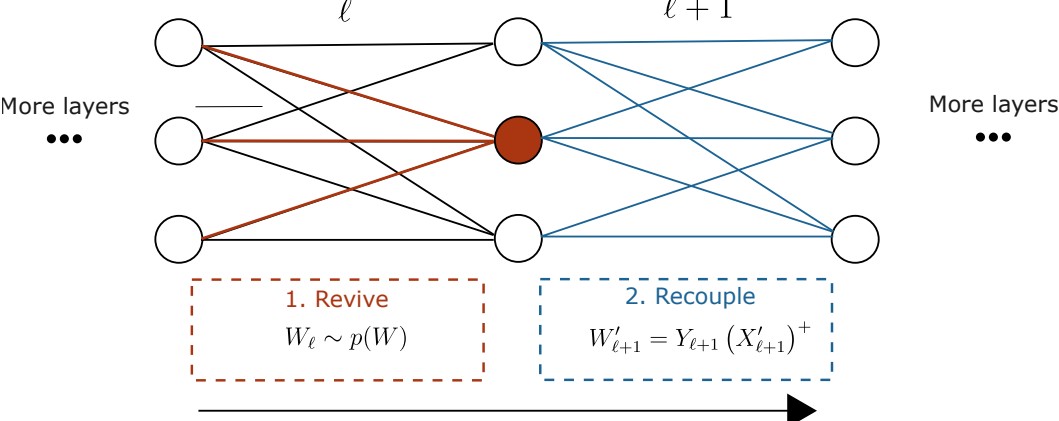

Figure 1: A schematic illustrating our method (RnR) for dense layers. For a dormant neuron we reset the incoming weights in the revive stage, causing a shift in the output distribution of the network. This is followed by a correction in the recouple stage in layer $\ell + 1$, where we solve a linear system to preserve the output distribution for all the parameters in the layer. This is done in a bottom-up fashion throughout the whole network.

## 1 INTRODUCTION

The ability of learning systems to continuously adapt and learn from new experiences within constantly shifting, non-stationary contexts is essential to the pursuit of general intelligence (Thrun, 1995; Zenke et al., 2017; Kirkpatrick et al., 2017; Berseth et al., 2022). A significant challenge to achieve this goal in neural networks is the progressive loss of their ability to learn from new experiences (Lyle et al., 2023). This phenomenon is commonly referred to as plasticity loss. This decline in learning ability has motivated extensive research, leading to numerous investigations into both its underlying mechanisms (Lyle et al., 2023; 2024) and mitigation strategies.

Continual Learning (CL) as a framework for tackling sequential decision-making problems is particularly susceptible to plasticity loss due to its constantly evolving tasks and datasets (Dohare et al., 2022). This can lead to pathological learning dynamics where the final performance of the network is severely compromised. However, the challenge is not confined to CL and extends to various machine learning paradigms where models encounter evolving training conditions, like Deep Reinforcement Learning (DRL), usually through changing target values or evolving replay buffers (Nikishin et al., 2022; D'Oro et al., 2023; Nikishin et al., 2023).

To address this critical issue, numerous approaches have been proposed. These strategies broadly encompass resampling network weights (Ash & Adams, 2020; Sokar et al., 2023; D'Oro et al., 2023; Nikishin et al., 2022; Dohare et al., 2022), architectural modifications (Nikishin et al., 2023; Abbas et al., 2023; Lyle et al., 2024; Liu et al., 2025a), and regularization techniques (Elsayed & Mahmood, 2024; Lewandowski et al., 2024; Kumar et al., 2023; Chung et al., 2024). While these methods have shown promise, they often provide only partial solutions, and suffer from significant drawbacks. In the case of parameter resampling, plasticity gains tend to come at the cost of significant performance loss, but are highly scalable, whereas others often come with signficant increase in the need for compute resources. Furthermore, these methods have been primarily designed for and evaluated on traditional dense network architectures.

In this work, we investigate the phenomenon on plasticity loss in transformer (Vaswani et al., 2017) architectures, whose plasticity dynamics in a CL setting remain largely unexplored. A system truly learns continuously when it not only retains past knowledge but also leverages it to efficiently learn new, related concepts. This ability to build upon past knowledge to learn new concepts is fundamentally stunted by a model's lack of plasticity. Through our experiments, we uncover unique challenges in network plasticity specific to transformers, which lead us to develop our own novel, efficient framework for mitigating plasticity loss. Specifically, our contributions include:

1. We find that existing mitigation strategies like ReDo (Sokar et al., 2023) and Shrink&Perturb (Ash & Adams, 2020) fail to mitigate plasticity loss in transformers. Recent approaches like ReGraMa (Liu et al., 2025b) do better, but at the cost of significant performance drops.

2. We find LayerNorm layers, which are known to help mitigate plasticity loss in dense networks (Lyle et al., 2023; 2024), also suffer from plasticity loss in transformers.

3. We propose a novel two-step framework Revive And Recouple (RnR) designed to increase network plasticity while retaining previous knowledge and evaluate its effectiveness.

The key idea behind RnR is to introduce a second step, Recouple, after a typical dormant neuron reset step (Sokar et al., 2023; Liu et al., 2025b), which we call Revive. Recouple serves the dual purpose of preserving the network output by solving a system of equations while enabling the newly revived neurons to participate in learning. A key advantage of the Recouple step is that it admits a closed-form solution.

We demonstrate the efficacy of RnR in mitigating plasticity loss in transformer architectures in a CL scenario with a growing, evolving dataset, where information learned from each CL stage helps downstream tasks.

## 2 RELATED WORK

Plasticity loss has motivated a great body of research with works exploring strategies to mitigate, prevent, identify and measure this phenomenon. We organize prior works in five broad categories and position RnR accordingly.

**Resampling network weights and resets.** Shrink and Perturb (Ash & Adams, 2020) shrink the network weights and perturb them with noise to rejuvenate the model, albeit with unpredictable effects on performance. Nikishin et al. (2022) reset the last few layers at regular intervals, at the cost of temporary performance drops. Further evolving this concept, D'Oro et al. (2023); Zhou et al. (2022) propose to fully reset the network at regular intervals. While they are very effective in mitigating plasticity loss, they result in a complete loss of performance immediately after the reset. These methods significantly alter the output of the network, while RnR preserves the network output as much as possible by design.

**Unit-level rejuvenation.** As opposed to resetting whole blocks of the network, these methods act on individual neuron units. Sokar et al. (2023) propose ReDo, which focuses on identifying dormant neurons based on a neuron activation based metric and resetting them. It is known to help mitigate pasticity loss in dense networks while also avoiding performance drops, but is known to suffer from sample efficiency issues. Along similar lines, Dohare et al. (2022) proposes Continual Backprop, an algorithm that keeps track of each neuron's utility in an online manner and re-initializes low utility neurons. Building upon ReDo, Liu et al. (2025b) propose ReGraMa which identifies dormant neurons using a gradient based metric. RnR lies in this category as the Revive step is typically any of the methods in this category. However, Recouple step makes it adjacent not completely fit in this category.

**Architectural changes.** Nikishin et al. (2023) introduce Plasticity Injection, which adds two additional heads to the network to inject plasticity. It perfectly retains the network output, but increases computation costs and lacks an efficient way to merge new networks. Abbas et al. (2023) show that using the CReLU activation function (Shang et al., 2016) helps mitigate plasticity loss. Lyle et al. (2023) propose adding layernorm layers before non-linearities in dense networks prevents plasticity loss. Building up on this, Lyle et al. (2024) also show that a combination of layernorm with weight decay is highly effective in mitigating plasticity loss. Lee et al. (2024) maintain two separate networks, one of which learns from the dataset(hare) and the other(tortoise) gets EMA-based updates to its parameters via the hare network.

**Regularization based methods.** Various regularization-based methods to prevent plasticity loss have also been proposed, such as Regenerative L2 (Kumar et al., 2023), Wasserstein regularization (Lewandowski et al., 2024), Weight Clipping (Elsayed & Mahmood, 2024), and Parseval regularization (Chung et al., 2024).

**Identifying and Measuring Plasticity Loss.** Identifying and measuring plasticity loss remains a complex challenge, as the underlying factors are not yet fully understood. Lyle et al. (2023) explore the factors influencing plasticity loss, examining the evolution of the loss landscape throughout training and proposes a metric to quantify plasticity loss. Further building upon this, Lyle et al. (2024) elaborate that plasticity loss occurs due to a combination of different conditions.

## 3 METHOD

Current approaches like Shrink and Perturb and ReDo perform well on MLPs, but fail to effectively mitigate plasticity loss on transformer based architectures since, as we shall see, dense layer resets are not enough to fix the plasticity issue in transformer architectures. Recent approaches like ReGraMa are able to effectively mitigate plasticity loss, but suffer from performance collapse since they rely on resetting of a large part of the network, thus lacking comprehensiveness. Here, we propose Revive And Recouple (RnR), a two-step framework to effectively increase network plasticity, while preserving the network output at little-to-no additional computation cost. The key idea behind RnR is to (i) restore plasticity in a layer by identifying dormant neurons and resetting their incoming weights, and (ii) an output preservation step in the next layer. This two-step procedure is performed sequentialy for all but the output layer of the network.

### 3.1 DESIDERATA FOR A ROBUST SOLUTION TO PLASTICITY LOSS

Addressing plasticity loss in CL requires a solution that is effective, practical, and scalable. Through our evaluation of existing approaches, we observe several limitations that hinder their overall effectiveness.

**Desiderata 3.1** (Plasticity Loss Mitigation)**.** Considering the limitations of existing approaches, we identify the following desiderata of a robust solution to plasticity loss:

> **(1) Scalability:** A robust solution should be scalable, i.e., it can be applied repeatedly without practical limitations.
>
> **(2) Output Retention:** The intervention should minimize or eliminate disruptions to the network's output.

**(3) Training Efficiency:** The solution must not introduce significant overhead that slows down training.

**(4) Sample Efficiency:** The solution should enable the agent to achieve high performance with fewer interactions with the environment.

As we shall see, we aim to fulfill the aforementioned desiderate by introducing a cheap correction step to the shifted outputs of the network layers after reviving dormant neurons.

## 3.2 NOTATION

For a feed-forward neural network with $L$ layers, the $\ell$-th layer has $H^\ell$ neurons and bias included parameters $W_\ell$. The network is trained on a data distribution $\mathcal{D}$. For a given input $x$, we denote the network loss as $\mathcal{L}(x)$. The output of the $i$-th neuron in the $\ell$-th is denoted as $h_i^\ell(x)$, and the corresponding pre-activation is denoted as $z_i^\ell(x)$.

A neuron is considered dormant if its contribution to learning diminishes, a condition characterized in some works by low activation magnitudes ($\mathbb{E}_{x\sim\mathcal{D}}|h_i^\ell(x)|$) or low gradient magnitudes ($\mathbb{E}_{x\sim\mathcal{D}}|\nabla_{z_i^\ell}\mathcal{L}(x)|$).

## 3.3 STAGE 1: REVIVE

A crucial step to mitigating plasticity loss is shifting the input of the neuron into a non-dormant regime that facilitates strong learning signals. Hence, the first stage of our method identifies dormant neurons and resamples their incoming weights. This detection is modular, and can be performed using various heuristics. In this work, we consider three such approaches: ReDo by Sokar et al. (2023), ReGraMa by Liu et al. (2025b), and a simple gradient based heuristic we name Low Gradient Quantile.

To adapt ReGraMa for use in transformers, we modify its formulation. The original method captures gradients with respect to the neuron's final output ($\nabla_{h_i^\ell}\mathcal{L}(x)$). However, this is suboptimal for transformers, because significant non-linear operations occur outside the feed-forward network like the Scaled Dot-Project Attention operation which can obscure the gradient signal isolated to an individual neuron. Therefore, we capture the gradients with respect to its pre-activation value ($\nabla_{z_i^\ell}\mathcal{L}(x)$). This modification has the key benefit of including the derivative of the activation function. Thus, accurately including the neurons which were rendered dormant by a vanishing activation derivative.

These heuristics assign a score to each neuron in layer $\ell$ and mark them dormant based on their respective criteria. We briefly discuss these heuristics below.

**ReDo** (Sokar et al., 2023) assigns an activation-based score as,

$$s_i^\ell = \frac{\mathbb{E}_{x\sim\mathcal{D}}\left|h_i^\ell(x)\right|}{\frac{1}{H^\ell}\sum_{k=1}^{H^\ell}\mathbb{E}_{x\sim\mathcal{D}}\left|h_k^\ell(x)\right|}.$$

**ReGraMa (modified)** assigns a gradient-based score as (original with $\nabla_{h_i^\ell}$ Liu et al. (2025b))

$$s_i^\ell = \frac{\mathbb{E}_{x\sim\mathcal{D}}\left|\nabla_{z_i^\ell}\mathcal{L}(x)\right|}{\frac{1}{H^\ell}\sum_{k=1}^{H^\ell}\mathbb{E}_{x\sim\mathcal{D}}\left|\nabla_{z_k^\ell}\mathcal{L}(x)\right|}.$$

**Low Gradient Quantile**: assigns a gradient-based score as

$$s_i^\ell = \mathbb{E}_{x\sim\mathcal{D}}\left|\nabla_{z_i^\ell}\mathcal{L}(x)\right|.$$

An important distinction between ReDo and ReGraMa is their generality. ReDo uses an activation scoring rule, which is only fitting for activations in which the gradient is close to $0$ when the activation is close to $0$. ReGraMa expands this by considering the gradient wrt. (pre-)activations of layer $l$.

Once the importance scores $s_i^\ell$ are assigned to the neurons, the methods differ in their criteria for identifying dormancy. ReDo and ReGraMa employ a fixed threshold $\tau$, classifying a neuron as dormant if $s_i^\ell \leq \tau$. We discard the normalization for the scoring and consider a ranking criterion in Low Gradient Quantile, identifying all neurons with scores below the $q$-quantile as dormant. The respective hyperparameters, $\tau$ and $q$ are determined empirically.

### 3.4 STAGE 2: RECOUPLE

In the revival stage, depending on the amount of dormant neurons identified by the dormancy criterion, the output of the network can shift significantly, leading to large performance drops. This is undesirable from the perspective of continual learning, where we want to preserve information learned and continue to improve. Hence we aim to introduce a correction to preserve the output distribution of the network. After the revival stage in layer $\ell$, the input to the layer $\ell + 1$ becomes $X'_{\ell+1}$. In this stage, we adjust its parameters to minimize the reconstruction loss with its pre-reset output $Y_{\ell+1}$. This keeps the change in layer output minimal while enabling the neurons we reset in the Revive stage to actively participate, acquiring stronger gradients. For the case of the transformer architecture, we identify two types of parameters that necessitate this correction and revival to encourage plasticity: the dense layer weights and layer norm parameters.

#### 3.4.1 LINEAR MODULES

For a linear module with weight $W_{\ell+1}$, pre-revive output $Y_{\ell+1}$, and post-revive input $X'_{\ell+1}$, we solve for the new weight $W'_{\ell+1}$ by solving the least-squares system

$$W'_{\ell+1} X'_{\ell+1} = Y_{\ell+1}.$$

This has the closed form solution using Moore-Penrose pseudo-inverse

$$W'_{\ell+1} = Y_{\ell+1} \left( X'_{\ell+1} \right)^+ .$$

Naturally, we cannot expect a perfect correction from solving the linear system, since 1) we only have access to a finite sample from the data distribution and 2) taking the pseudo-inverse. Nevertheless, as we shall see in Section 4, providing this correction significantly reduces performance drops in comparison to other methods.

#### 3.4.2 LAYERNORM MODULES

For a LayerNorm module with affine params $\gamma_{\ell+1}, \beta_{\ell+1}$, dropping the layer number $\ell + 1$ for brevity to get $\gamma, \beta$, pre-revive output $Y$, and post-revive input $X'$. The normalized inputs to the affine params become $\hat{X}'$. We solve for the new affine params $\gamma', \beta'$ per-feature using the following system for $i$-th feature:

$$\gamma'_i \hat{X}'_i + \beta'_i = Y_i,$$

which gives us the closed form solutions

$$\gamma'_i = \frac{\mathrm{Cov}\left( \hat{X}'_i, Y_i \right)}{\mathrm{Var}\left( \hat{X}'_i \right)}, \qquad \beta'_i = \mathbb{E}\left[ Y_i \right] - \gamma'_i \mathbb{E}\left[ \hat{X}'_i \right].$$

### 3.5 IMPLEMENTATION DETAILS

We use a larger representative batch $\mathcal{B}$ to calculate expected values and pseudo-inverse. It is also important to note that $X'_{\ell+1} \neq W'_l X'_l$ due to either transformer-specific operations that do not involve feed-forward layers or due to activation functions. This two-step procedure is applied iteratively to all the layers of the network except the last layer. We provide more details on how RnR is applied in practice in Algorithm 1.

## 4 EXPERIMENTS

We evaluate the effectiveness of RnR on the Continual Learning scenario from Lee et al. (2024), which simulates an evolving dataset with increasing size and decreasing levels of noise. The data

**Algorithm 1** RnR Pseudocode

**Require:** `M`: network, `r`: reset interval
**Require:** `o_layers`: ordered list of layers with trainable params.
```
 1: for step,(x,y) in enumerate(loader) do
 2:     logits = M(x)
 3:     loss = loss_fn(logits, y)
 4:     loss.backward()
 5:     optimizer.step()
 6:     if step%r==0 then
 7:         B ← Sample a large representative batch from D
 8:         for i in range(len(o_layers)-1) do
 9:             Identify dormant neurons in o_layer[i] using B (see Section 3.3)
10:             Re-initialize params for dormant neurons
11:             Update params for o_layer[i+1] (see Section 3.4)
12:         end for
13:     end if
14: end for
```

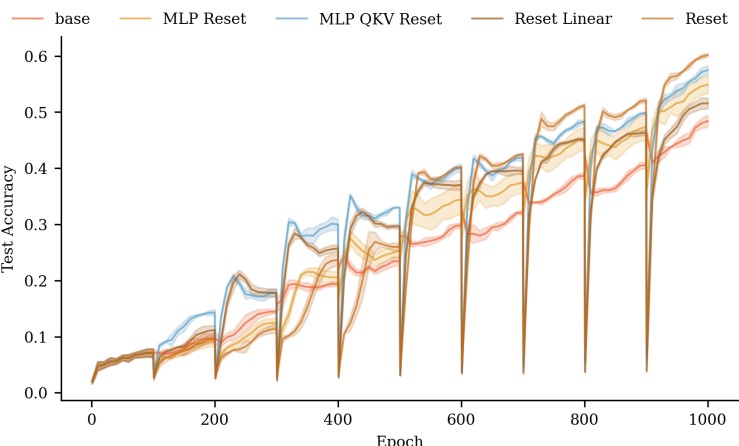

Figure 2: Investigating the effect of resets on training ViT-Tiny on CIFAR100. For each line we reset different parts of the network when the tasks change. Even if we reset all the linear layers, we are not able to match the final performance of *Resets*, hinting to the presence of plasticity loss in LayerNorm layers.

is divided randomly into 10 chunks which are then used to create datasets for the CL tasks. The CL scenario consists of 10 tasks, with the corresponding dataset being augmented by a new chunk every task. Additionally, label noise is introduced by randomly relabeling a fraction of the data. This noise ratio is descreased with tasks reducing from 0.5 to 0.0. Consequently, the last task in the scenario has access to the full dataset and does not contain any noise. We run this scenario in two settings, (i)ViT-Tiny on CIFAR100, and (ii)ViT-Small on Tiny Imagenet. We set the task budget to 100 epochs for ViT-Tiny and 40 epochs for ViT-Small, evaluating every 10 and 5 epochs respectively. Additionally, we also evaluate one epoch after intervention to record the drop in performance and the subsequent recovery speed. We also separately evaluate right after the intervention to calculate performance drops. We use a representative batch $B$ of size 2048 to calculate expected values in Section 3. A detailed list of hyperparameters can be found in Supplementary A.

We conduct an ablation study that resets various components of the ViT network to comprehend why earlier techniques such as ReDo and Shrink & Perturb, which have successfully alleviated plasticity loss in conventional dense networks, do not work with transformers. We train ViT-Tiny on the CL setting described above using the CIFAR100 dataset as reported below.

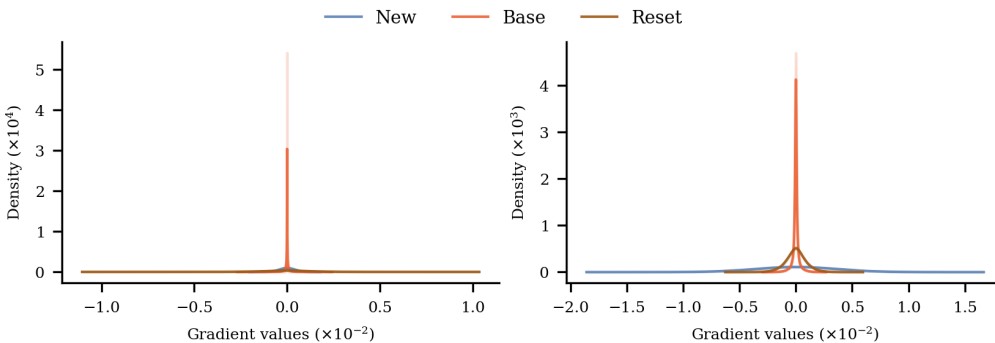

Figure 3: Distribution of average incoming gradients to the neurons in the LayerNorm preceding the final MLP block (**left**) and the LayerNorm before the classification head (**right**) in ViT-Tiny, after being trained on the CL scenario (Figure 2). Gradients for the baseline are highly concentrated around 0 compared to the case with full resets and to an newly initialized network, clearly showing loss of plasticity in LayerNorm. (3 seeds, mean±std)

### 4.1 LayerNorm and Plasticity Loss

Figure 2 shows the performance of ViT-Tiny on CIFAR100 on the CL scenario. We incrementally reset different parts of the network to identify the parts that suffer from plasticity loss. We start with resetting the MLP blocks as most work studying plasticity loss has been done on dense networks. Resetting QKV weight parameters along with MLP blocks results in significant performance gains. Interestingly, resetting all the fully connected layers in the network performs worse than resetting the MLP blocks and QKV layers. We hypothesize that this is due to the loss of meaningful learned representations from the previous tasks. However, the most significant insight from this ablation study is that even after resetting all fully-connected layers, the network still suffers from plasticity loss, not being able to reach the performance of full resets. This indicates that LayerNorm layers, which have proven effective in mitigating plasticity loss in dense networks (Lyle et al., 2023; 2024), also suffer from plasticity loss in transformer architectures.

To gain further insights, we take final checkpoints from the CL runs on ViT-Tiny for base and resets, pass the whole dataset through them to gather average incoming gradients $\nabla_{z_i^\ell} \mathcal{L}$ (averaged over batch and context length) for each neuron per layer. We then plot the distributions for these average gradient values and compare the differences between 3 checkpoints: (i) base, (ii) reset, (iii) newly-initialized network. Figure 3 presents this distribution for the final two LayerNorm layers in the network. We show the deeper layers as other works have shown deeper layers to have a higher degree of plasticity loss (Nikishin et al., 2022). We find that for the baseline checkpoint with plasticity loss, incoming gradients are highly concentrated around 0 compared to the reset or a newly trained checkpoint. This is a definitive proof that layer norms contribute to plasticity loss, although they have been shown to effectively reduce it in non-transformer networks. Thus, mitigation strategies need to take layer normalization into account.

### 4.2 Continual Learning

Figure 4 showcases the effectiveness of RnR compared to Shrink & Perturb, ReDo, and resets in mitigating plasticity loss on ViT-Tiny and ViT-Small. While Shrink & Perturb and ReDo are able to preserve network output leading to performance drops close to 0, they are not able to increase network plasticity in both cases. Resets are able to achieve the highest performance, but have the well-established pitfalls of complete performance collapse, throwing away crucial knowledge from previous tasks which makes them relatively slower to regain performance. RnR effectively mitigates plasticity loss in both cases and achieves performance comparable to resets. Moreover, while it is not able to preserve the output as well as Shrink & Perturb and ReDo, it is significantly superior to resets. We tried all 3 strategies for Revive step (Section 3.3) in both settings and plot our best runs in Figure 4. For ViT-Tiny, our simple approach based on Low Gradient Quantile with a threshold value of 0.4 performed the best whereas on Vit-Small, ReGraMa with a threshold value of 1.0 had the highest final performance, albeit at the cost of slightly larger performance drops. We argue that RnR

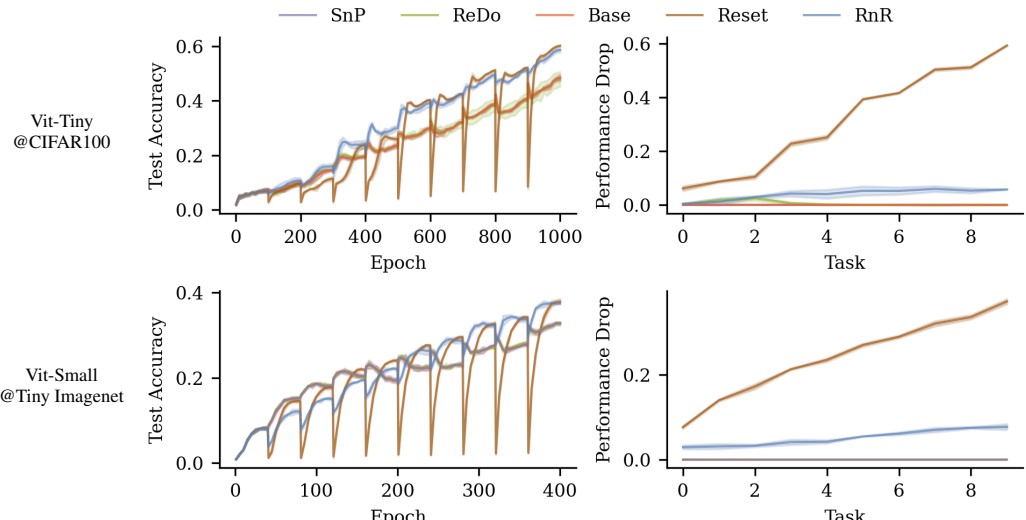

Figure 4: Performance of RnR in the CL scenario. **Top:** ViT-Tiny on CIFAR100. RnR is using Low Quantile Gradient with $\tau = 0.4$. **Bottom:** ViT-Small on Tiny Imagenet. RnR is using ReGraMa with $\tau = 1.0$. For both rows, **left** shows that RnR successfully mitigates plasticity loss, achieving a final performance comparable to resets, while the **right** plot shows the immediate performance drop from the intervention, which RnR drastically reduces. All results with 3 seeds, mean±std.

provides an effective framework that is able to achieve performances comparable to full resets while preserving crucial knowledge from previous tasks. These preserved learned representations from previous tasks result in faster recovery from performance drops compared to resets. Furthermore, it does not add significant training overhead, thus satisfying all desiderata from Desiderata 3.1.

### 4.3 HOW TO REVIVE?

One of the most crucial decisions for the success of RnR is the choice of heuristic for the Revive step (Section 3.3). This step is directly responsible for increasing network plasticity by resetting dormant neurons. Therefore, it is highly important to make the right choice. For this purpose, we compare the performance of ReDo, ReGraMa, and Low Gradient Quantile heuristics as defined in Section 3.3 for ViT-Tiny and ViT-Small in Section 4.3. We test different threshold values for all of the methods and

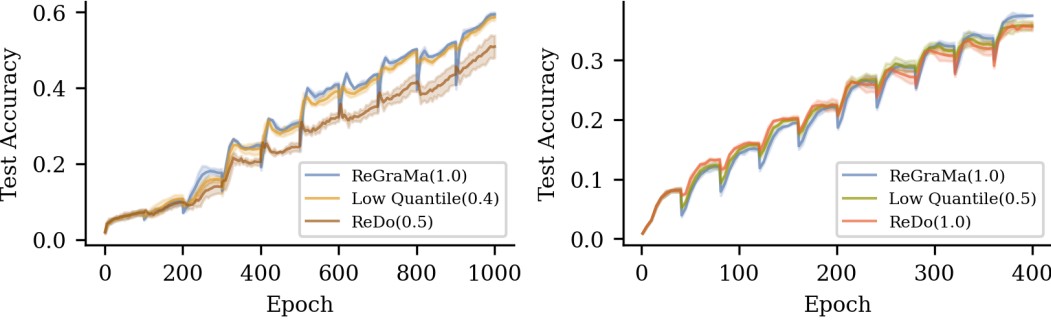

Figure 5: Comparison of different techniques for the Revive step in RnR on ViT-Tiny (**left**) and ViT-Small (**right**). Thesholds are specified in parenthesis. ReGraMa with a threshold of 1.0 typically achieves higher final performance, while Low Gradient Quantile results in smaller performance drops. All results with 3 seeds ± std.

compare the best runs for each strategy in both models as shown in Figure 5. We find that ReGraMa is able to achieve the best final performance compared to ReDo and Low Gradient Quantile methods, albeit at a significantly higher threshold than the original work's suggested value of 0.01 (however,

notice that we changed the heuristic to pre-activation instead of post-activation gradients). Low Gradient Quantile on the other hand, comes close to ReGraMa in terms of final performance, but lags slightly behind. However, in terms of performance drops, Low Gradient Quantile performs better. ReDo is not able to perform well on ViT-Tiny but does show decent performance on ViT-Small. Thus, we evaluate the trade-off and pick the suitable metric. Future work is needed to test these heuristics on other architectures and settings to converge to a general recommendation.

## 5 CONCLUSION AND FUTURE WORK

In this paper, investigate plasticity loss in transformer architectures in continual learning (CL) settings. We show that plasticity loss mitigation strategies designed for dense networks do not work on transformer architectures in CL scenarios with an evolving dataset. We provide evidence that LayerNorm layers also contribute to plasticity loss, albeit previously known to help mitigate it in dense architectures. We present a novel two-step framework Revive And Recouple (RnR) which effectively mitigates plasticity loss in transformers by first increasing network plasticity and then recouples to preserve output and make newly reset neurons participate in learning. Finally, we compare a few pre-existing heuristics, ReDo and ReGraMa along with our own Low Gradient Quantile approach in terms of how they perform within our framework. As RnR depends on the choice of heuristics in the Revive step, further work can be done towards accurately identifying dormant neurons to improve the overall efficacy. Furthermore, since the framework is generic, we believe it can potentially be applied to other data-rich settings where plasticity loss is a known problem like off-policy Reinforcement Learning. In the future, we would also like to examine RnR's efficacy in larger scale transformer architectures.

### DECLARATION OF LLM USAGE

We acknowledge the use of Large Language Models as an auxiliary tool in the preparation of this manuscript. The models' application was limited to proofreading and grammatical checks, and vocabulary support to enhance prose clarity. Additionally, LLMs were used to generate boilerplate code for visualizations. All LLM-generated outputs were critically reviewed, revised, and validated for correctness by the authors, who take full responsibility for the final content of this paper.

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

## A  TRAINING DETAILS

This section contains experimental details and hyperparameters for reproduceability.

### A.1  DATASETS

We use 2 datasets throughout our work:

- **CIFAR100** (Krizhevsky, 2009): Consists of 60,000 color images (50,000 for training and 10,000 for test) of size $32 \times 32$ pixels distributed uniformly over 100 classes.
- **Tiny Imagenet**: A scaled down subset of the full ImageNet dataset (Russakovsky et al., 2015). Consists of 110,000 color images(100,000 for training and 10,000 for validation set), scaled down from original ImageNet to size $64 \times 64$ pixels distributed uniformly over 200 classes.

### A.2  ARCHITECTURES

We use ViT-Tiny and ViT-Small models for our study. The architectural details are as follows:

- **ViT-Tiny**: Trained on CIFAR100. Following the setup from Lee et al. (2024) It had patches of size 4x4, embedding dimension of 192, 3 attention heads, and 12 transformer blocks. We used a learning rate of 0.003, AdamW as optimizer, weight decay of 0.05, layer normalization, and a dropout rate of 0.1.
- **ViT-Small**: Trained on Tiny Imagenet. This is a larger version of ViT-Tiny but follows the same architecture underneath. It had patches of size $8 \times 8$, embedding dimension of 384, 6 attention heads, and 12 transformer blocks. We used a learning rate of 0.0001, AdamW as optimizer, weight decay of 0.05, layer normalization, and a dropout rate of 0.1.

### A.3  HYPERPARAMETER DETAILS

In this section, we provide hyperparameter details using our experiments for reproduceability:

| Hyperparam | Value |
|---|---|
| Optimizer | AdamW |
| Optimizer Hyperparams($\beta_1, \beta_2$) | (0.9, 0.999) |
| Batch Size | 256 |
| Learning Rate Scheduler | Warmup+Cosine Annealing |
| Warmup Ratio | 0.1 |
| Initial Learning Rate | 0 |
| Gradient Clip | 0.5 |

Table 1: Hyperparams for training

For our baselines, we use $\tau = 0.1$ for ReDo, and $\alpha = 0.8$ for Shrink and Perturb

