# OpenReview forum: "Revive and Recouple: Mitigating Plasticity Loss in Transformer Architectures"
_ICLR.cc/2026/Conference — Submitted to ICLR 2026_

### Official Review · Reviewer_q5mA · 2025-10-31

**Soundness:** 1
**Presentation:** 1
**Contribution:** 3
**Rating:** 2
**Confidence:** 5

**Summary:**

The paper studies loss of plasticity in transformer architectures. The results show a substantial loss of plasticity in certain Vision Transformers. The analysis shows that layer normalization also contributes to loss of plasticity in these architectures, even though LN has been shown to reduce loss of plasticity in feed-forward networks. The final step is to propose a new strategy, Revive and Recouple (RnR), that reduces disruption in the network output when a large number of units are reset.

**Strengths:**

The clear demonstration that layer normalization is contributing to loss of plasticity (in Figure 3) is novel and useful. The recouple strategy mitigates a significant issue with algorithms like ReDo and Regrama, making unit-resetting algorithms more practical.

**Weaknesses:**

Although the paper presents some useful ideas, it is poorly written and presented, a lot of relevant literature is missing, the results are not statistically significant, and some baselines are missing.

* Presentation issues. There are several presentation issues in the paper. First, the colours in Figure 2 are all just different shades of beige/brown. It is not legible. One of the labels is "reset linear", but it is never explained what it refers to or how it differs from "reset MLP". When discussing Figure 2 in the first paragraph of section 4.1, it says "resetting all the fully connected layers ..." What exactly is "resetting all the fully connected layers"? Is this the "reset linear" label? If so, how is it different from "reset MLP"? Similar issues are present in Figure 3. What is the difference between "reset" and "newly-initialized network"? When exactly is the quantity plotted in Figure 3 measured, and for which layers of the network? Similar issues are present in the text. Desiderate 3.1 on line 159 is called "scalability", but what is described is "it can be applied repeatedly without practical limitations." I don't think this is what scalability means; it seems a better phrase for description is "ease of use".

* Missing prior work. The paper claims that plasticity loss in transformers is largely unexplored. This is somewhat true, as recent papers have explored plasticity loss in transformers. Specifically, Farias and Jozefiak (2024) studied plasticity loss in transformers, and Hernandez-Garcia et al. (2025) studied plasticity loss in vision transforms, very similar to the architectures presented in this paper. These recent papers should be discussed in detail in this paper.

* Missing baselines: The paper claims that unit resetting algorithms can dramatically change the output of the network. That is true for algorithms like ReDo and ReGrama, but not for continual backpropagation, as it typically only resets one unit per layer at a time. However, there is no comparison presented, and this difference is not even acknowledged in the paper.

* Statistical significance. The experiments only show the results for three seeds and plot the standard deviation. Standard deviation is not a measure of statistical significance. Standard error or bootstrapped confidence interval should be plotted. Additionally, all the experiments should include 30 runs for each algorithm. Hernandez-Garcia et al. (2025) showed that Vision transformers using ReDo or continual backpropagation can collapse in class-incremental problems. However, this only happens once every 20 to 30 runs. Conducting just 3, or even 10 runs, strongly reduces the chances of seeing behaviours like this. Does RnR mitigate this collapse behaviour?

Farias and Jozefiak, SELF-NORMALIZED RESETS FOR PLASTICITY IN
CONTINUAL LEARNING, ICLR 2025.
Hernandez-Garcia et al. REINITIALIZING WEIGHTS VS UNITS FOR MAINTAINING PLASTICITY IN NEURAL NETWORKS, CoLLAs 2025.

**Questions:**

1. What hyperparameters were tried for Redo and SnP? Please include that detail in the appendix and perform an appropriately wide parameter sweep for all algorithms.
2. What is the added computational cost of the recouple step? Please provide the additional wall clock time for one of the experiments.
3. Have you tried RnR on feed-forward networks? Does it improve ReDo or ReGrama? It would be a good idea to conduct a toy experiment on a problem like random label MNIST.

---

### Official Review · Reviewer_Mfid · 2025-10-31

**Soundness:** 1
**Presentation:** 3
**Contribution:** 2
**Rating:** 4
**Confidence:** 4

**Summary:**

The authors propose Revive and Recouple (RnR), a two-step framework to mitigate plasticity loss in transformer architectures. RnR first “revives” dormant neurons by resetting their weights, then “recouples” downstream layers through a closed-form correction to preserve the network’s output distribution. Experiments on ViT-Tiny and ViT-Small show RnR improves adaptability over existing reset-based methods while retaining prior knowledge.

**Strengths:**

- The paper tackles an important and under-explored issue — plasticity loss in transformer architectures.

- The proposed RnR framework is a neat idea. The first step (Revive) reinitializes dormant neurons, and the second (Recouple) adjusts downstream weights in a closed-form way to preserve the model’s output distribution.

- The paper is written clearly overall, and was easy to follow.

**Weaknesses:**

- I think the main weakness is the narrow empirical scope. All experiments are done on ViT-Tiny and ViT-Small using CIFAR-100 and Tiny-ImageNet — both small-scale. This makes it hard to know if RnR scales to realistic transformer workloads (like ViT-Base or LLMs). The method is motivated as a general solution, but the validation is quite limited.

- Missing comparisons. The authors cite recent works (such as Continual Backprop, Hare–Tortoise) but don’t actually compare against them which is relevant baselines.

- Hyperparameter and robustness analysis is missing. It remains unclear how sensitive RnR is to choices like reset interval, threshold τ, or batch size. Even parameters like learning rate and weight decay can drastically affect plasticity, but there’s no mention of tuning or ablation.

- The pseudo-inverse step can be expensive, especially with large representative batches. A proper runtime or memory analysis is missing.

- Maybe I misunderstood soemthing here - the authors claimed that RnR preserves network outputs while restoring plasticity. However, Figure 4 and Section 4.2 show the opposite—RnR does not preserve outputs as well as Shrink & Perturb or ReDo. In practice, it trades some output stability for faster recovery and higher final performance.

- The authors stated that prior methods were only studied on dense networks, but several cited works have also been evaluated on ViTs or transformer-based architectures. Additionally, testing (or at least discussing) RnR on MLPs or CNNs could help demonstrate that the approach generalizes beyond ViTs.

- I also think the claim that *LayerNorm layers themselves suffer from plasticity loss* is a bit overstated. The results show reduced gradients, but that doesn’t necessarily prove causation. Prior work (e.g., Lyle et al., 2023; 2024) showed LayerNorm tends to reduce plasticity loss in dense networks, not that it eliminates it.

**Questions:**

Please refer to the comments in the weaknesses section.

---

### Official Review · Reviewer_RVo2 · 2025-11-01

**Soundness:** 1
**Presentation:** 3
**Contribution:** 1
**Rating:** 2
**Confidence:** 4

**Summary:**

This paper tackles plasticity loss specifically in Transformer architectures. The authors show that Transformers suffer from plasticity loss not only in the MLP layers but also in the LayerNorm parameters. To address this issue, the authors propose Revive and Recouple (RnR), a two-step framework that restores network plasticity while preserving previously learned knowledge, thereby avoiding the performance collapse that is typical in existing approaches.

**Strengths:**

* The proposed method, RnR, outperforms existing approaches while exhibiting a smaller performance drop than full resetting.
* The paper introduces a novel “recouple” mechanism that minimizes performance drop after reset.
* The paper is well written and easy to follow.

**Weaknesses:**

The paper lacks novelty.
* It has been shown that methods designed to mitigate trainability loss (such as ReDo) are ineffective in mitigating generalizability loss [1]. Thus, it is not surprising that ReDo shows poor performance in the setting presented in Figure 4.
* Previous works have already demonstrated that plasticity loss exists in Transformer architectures [1, 2, 3].
* While the aim of this paper is to tackle the loss of plasticity, the proposed method employs existing approaches from ReDo and ReGraMa to mitigate plasticity loss.

The paper does not provide appropriate and sufficient evidence to support its claims.
* The authors argue that the LayerNorm layer suffers from plasticity loss by showing that resetting only fully connected layers cannot match the performance of full resets. However, this indicates that the learnable parameters in LayerNorm contribute to plasticity loss, not the normalization itself. To prove that LayerNorm suffers from plasticity loss, the authors should perform the same comparison without learnable parameters in LayerNorm.
* While the authors mention that ReGraMa shows significant performance drops, the corresponding result is not presented in Figure 4.
* Why not simply perform additional gradient updates on the representative batch to recover knowledge? The authors should provide justification for performing iterative correction instead of directly applying additional gradient updates on the representative batch (e.g., superiority in computational cost or effectiveness). Without such comparative evaluation, it is impossible to judge the method’s effectiveness and practicality.
* The authors present desiderata for their method, but the supporting evidence for the scalability, training efficiency, and sample efficiency of RnR is not presented in the experimental section.
* The experimental settings are insufficient. The authors should consider more diverse settings, such as warm-start [1] or class-incremental settings [4, 5].

The results in the paper contradict previous work.
* According to the results from [1], Shrink & Perturb mitigates plasticity loss in Transformer architectures (see Figure 6, ViT-Tiny with CIFAR-100). The authors mentioned that they used the same setting as [1], but their results contradict these findings.

[1] Lee, Hojoon, et al. "Slow and steady wins the race: Maintaining plasticity with hare and tortoise networks." *arXiv preprint arXiv:2406.02596* (2024).
[2] Lyle, Clare, et al. "Understanding plasticity in neural networks." *International Conference on Machine Learning*. PMLR, 2023.
[3] Lyle, Clare, et al. "What Can Grokking Teach Us About Learning Under Nonstationarity?." *arXiv preprint arXiv:2507.20057* (2025).
[4] Lewandowski, Alex, et al. "Learning continually by spectral regularization." *arXiv preprint arXiv:2406.06811* (2024).
[5] Park, Sangyeon, et al. "Activation by interval-wise dropout: A simple way to prevent neural networks from plasticity loss." *arXiv preprint arXiv:2502.01342* (2025).

**Questions:**

How is the size of the representative batch chosen? How sensitive is RnR to the size of the representative batch?

---

### Official Review · Reviewer_Tsdk · 2025-11-05

**Soundness:** 2
**Presentation:** 2
**Contribution:** 2
**Rating:** 2
**Confidence:** 4

**Summary:**

This work explores a reset based plasticity loss mitigation technique for Transformer based architectures. Specifically, they propose a slightly different criterion for selecting neurons to reset and they adjust the weights of the layer after the reset layer to achieve the same output distribution of activations as pre-reset. They evaluate their approach on a continual supervised learning setup where the data is broken into 10 chunks and a chunk is added (with a decreasing amount of label noise for each chunk) for each task.

**Strengths:**

-The idea that layer norm parameters are what are causing issues in reset based plasticity methods is interesting and novel.
- The method suggested is intuitive and the story for it makes sense, although I think more work needs to be done for empirical justification.

**Weaknesses:**

- A big part of the story of the paper is that resetting weights with current approaches creates some kind of distribution shift in the activations that needs to be corrected. I think this needs to be more thoroughly shown. First, the existence of the shift, and how severe it is. Second, an exploration of how it causes issues.
- I think you need additional evaluations. More of the settings presented in Lee et al. (2024) such as the warm start settings, but also settings such as RL should be added.
- I think another ablation could be added exploring whether applying RnR to the whole network is necessary, or whether just applying it to certain parts of the network is enough. A similar experiment to Figure 2.
- I expand on this in the Questions section, but I am not convinced that all the results are correct, given there's no difference between ReDo, Base, and SnP in the main experiments.

**Questions:**

- Could you clarify what the difference between MLP-QKV and Reset Linear is? Also is the difference between Reset Linear and Reset the additional reset of the Layernorm parameters?
- When training with your method, is there any notion of convergence? Am I correct in thinking that your method will never actually converge, since you always reset some proportion of the network?
- For resetting the linear modules, isn’t the solution to maintain the output distribution simply to zero out the weights connected to the reset neurons? If so, how is that different than ReDo?
- In the original Lee et al. (2024) where the setting was introduced, SnP was a very strong baseline, achieving close to the best performance on the setting that you are using (over 50% accuracy at the end of continual training on CIFAR-100 with ViT-Tiny, which matches your baseline). Here, it’s a bit unclear, but it seems like SnP’s performance is essentially the same as base (40%). There doesn’t seem to be a performance drop with SnP, but there was in the Lee et al paper. In general, there doesn’t seem to be a difference at all between Base, ReDo, and SnP, which seems a bit odd. Could you explain the difference?

---

### Meta-Review · Area_Chair_Reb1 · 2026-01-02

**Summary:**

The reviewers unanimously identified shortcomings in the paper, primarily focusing on the lack of novelty and the restriction to small-scale architectures and datasets. Several concerns are raised regarding contradictions with prior literature, particularly concerning the performance of baselines like Shrink & Perturb and the claims about LayerNorm's role in plasticity loss. Additionally, reviewers also mentioned the lack of discussion and comparison with relevant baselines such as Continual Backprop and various presentation issues. The insight behind the recouple step and its computational cost are also questioned.

**Reviewer Concerns:**

The authors did not submit a rebuttal or engage in the discussion. Hence, none of the issues raised by the reviewers were addressed.

**Reviewer Scores:**

Since there is no response to the comments from reviewers, reviewers would maintain their original scores.

---

### Decision · Program_Chairs · 2026-01-26

Reject